# Traveling Words: A Geometric Interpretation of Transformers

## Abstract

Transformers have significantly advanced the field of natural language processing, but comprehending their internal mechanisms remains a challenge. In this paper, we introduce a novel geometric perspective that elucidates the inner mechanisms of transformer operations. Our primary contribution is illustrating how layer normalization confines the latent features of the transformer to a hypersphere, subsequently enabling attention to mold the semantic representation of words on this surface. This geometric viewpoint seamlessly connects established properties such as iterative refinement and contextual embeddings. We validate our insights by probing a pre-trained 124M parameter GPT-2 model. Our findings reveal clear query-key attention patterns in early layers and build upon prior observations regarding the subject-specific nature of attention heads at deeper layers. Harnessing these geometric insights, we present an intuitive understanding of transformers, depicting iterative refinement as a process that models the trajectory of word particles along the surface of a hyper-sphere.

## 1 Introduction

The transformer architecture (Vaswani et al., 2017) has sparked a significant shift in Artificial Intelligence (AI). It is the central component behind some of the most advanced conversational AI systems (Brown et al., 2020; Thoppilan et al., 2022; Bai et al., 2022), and has been established as state-of-the-art for Natural Language Processing (NLP), Computer Vision (CV) and Robotics applications, and many other tasks (OpenAI, 2023; Google, 2023; Chen et al., 2023; Zong et al., 2022; Driess et al., 2023).

Recent work on the interpretability of the transformer architecture has focused on analyzing weights in relation to the word embedding space used in its input and output layers Dar et al. (2022); Elhage et al. (2021); Geva et al. (2022); Brody et al. (2023); Windsor (2022); Millidge & Black (2022). Elhage et al. (2021) introduces "Transformer Circuits", a theoretical framework that decomposes the transformer computation into two main components: a residual stream that carries information from input to output layers and attention/feed-forward updates that modify the information

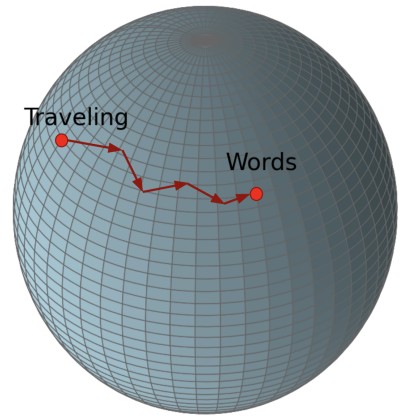

Figure 1: Overview of the proposed geometric interpretation of Transformers. The input token "Traveling " is embedded as a word particle onto a hyper-sphere, and residual updates determine the path that the particle will follow along the surface, culminating on the region closest to the next token: "Words".

flowing in the residual stream. A key development from their work is grouping attention matrices into the virtual interaction matrices $W_{QK}$ and $W_{OV}$, exploring their role in updating the information carried throughout the transformer's residual stream. Geva et al. (2022) demonstrate that the updates from the feed-forward module can be decomposed into a linear combination of sub-updates

given by the weight matrix of the feed-forward module's second layer. This matrix directly interacts with the residual stream and allows the authors to measure the impact of each sub-update on the model's final prediction using the matrix $W_E$ as a probe. Dar et al. (2022) incorporate these ideas to show that it is not only possible to interpret the outcomes of each transformer operation in relation to its latent space but also the weights themselves, enabling them to do zero-shot model stitching by "translating" between latent spaces of different language models. Finally, Millidge & Black (2022) note that analysis on the singular vectors of the $W_{OV}$ matrix provides better practical results when compared to analysis of its row and column weights.

A complimentary perspective to the line of work on Transformer Circuits comes from the geometric interpretation of layer normalization (Ba et al., 2016) by Brody et al. (2023). The authors prove that layer normalization is equivalent to projecting features onto the hyperplane defined by the $\overrightarrow{1}$ vector and then scaling the projection by $\sqrt{d}$. They show that these properties are crucial for the attention mechanism to either attend to all keys equally or to avoid the problem of having "unselectable" keys (relevant keys within the convex hull of a set of non-relevant keys). The study by Windsor (2022) offers additional evidence for the representational power of layer normalization, visualizing the highly non-linear behavior resulting from this operation and demonstrating that, when employed as an activation function in a neural network, layer normalization can solve complex classification tasks.

In this work, we build upon these two perspectives to propose a novel geometric interpretation of transformers. In subsection 2.1, we introduce an alternative equation for layer normalization based on its geometric properties. In subsection 2.2 and subsection 2.3, we discuss the implications of this equation on the attention module, its impact on the transformer's output probabilities and how it relates to the concept of iterative refinement. Finally, we provide results on our probing experiments in section 3, demonstrating the benefits of our approach on interpretability. An illustrated summary of the proposed geometric interpretation is given in Figure 1.

## 2 TRANSFORMERS AS A COMPOSITION OF GEOMETRIC PRIMITIVES

In this section, we analyze each of the transformer's components from a geometric perspective, leveraging the interpretation of one component to analyze the next. We begin with the layer normalization function, for which we demonstrate that it constrains $d$-dimensional input features to lie within the surface of a $(d-1)$ dimensional hyper-sphere. Then we consider the role of the $W_{QK}$ matrix in terms of geometric transformations on said hyper-sphere, and the $W_{VO}$ matrix as a key-value mapping from the hyper-sphere back to $\mathbb{R}^d$, highlighting its similarities with the key-value interpretation of the feed-forward module proposed by Geva et al. (2021). Finally, we discuss the role of the embedding matrix $W_E$ on the transformer's output probabilities.

### 2.1 LAYER NORMALIZATION

In its original formulation (Ba et al., 2016), layer normalization is introduced using the mean $\mu$ and standard deviation $\sigma$ computed along the dimensions of an input vector $x \in \mathbb{R}^d$:

$$\text{LayerNorm}(\boldsymbol{x}) = \frac{\boldsymbol{x} - \mu}{\sigma} \tag{1}$$

However, recent work by Brody et al. (2023) presents an alternate perspective on layer normalization, interpreting it as the composition of two distinct geometric transformations. The difference $\boldsymbol{x} - \mu$ is shown to be orthogonal to the $\overrightarrow{1}$ vector, suggesting that the features of the input vector are projected onto a hyperplane $\mathcal{H}$ defined by the normal vector $\overrightarrow{1}$, and the denominator $\sigma$ is formulated in terms of the norm of the projected vector as follows:

$$\sigma = \frac{1}{\sqrt{d}}||\boldsymbol{x} - \boldsymbol{\mu}|| \tag{2}$$

Building on this insight, we demonstrate (Appendix A) that the mean $\mu$ is the projection of $x$ onto the vector $\frac{1}{\sqrt{d}}\overrightarrow{1}$, as opposed to directly onto $\overrightarrow{1}$. This finding simplifies the computation of the projection of $x$ onto the hyperplane $\mathcal{H}$:

$$\text{proj}_{\mathcal{H}}(\boldsymbol{x}) = \boldsymbol{x} - \text{proj}(\boldsymbol{x}, \frac{1}{\sqrt{d}} \overrightarrow{\boldsymbol{1}})$$

$$= \boldsymbol{x} - \boldsymbol{\mu} \tag{3}$$

Incorporating Equation 2 and 3 into Equation 1, we obtain a geometric formula for layer normalization:

$$\text{LayerNorm}(\boldsymbol{x}) = \sqrt{d} \, \frac{\text{proj}_{\mathcal{H}}(\boldsymbol{x})}{||\text{proj}_{\mathcal{H}}(\boldsymbol{x})||_2} \tag{4}$$

Intuitively, layer normalization projects a vector $\boldsymbol{x} \in \mathbb{R}^d$ to the hyperplane $\mathcal{H}$ perpendicular to $\frac{1}{\sqrt{d}} \overrightarrow{\boldsymbol{1}} \in \mathbb{R}^d$, and normalizes the projection such that it lies on the surface of a $d-1$ dimensional hyper-sphere of radius $\sqrt{d}$ (for a visual understanding of this transformation with $d = 3$, refer to Figure 2). Furthermore, layer normalization typically incorporates a scaling factor $\gamma$ and a bias term $\boldsymbol{\beta}$. The scaling factor $\gamma$ acts along each coordinate axis, transforming the hyper-sphere into a hyper-ellipsoid, while the bias term $\boldsymbol{\beta}$ translates the ellipsoid's center away from the origin (Figure 3).

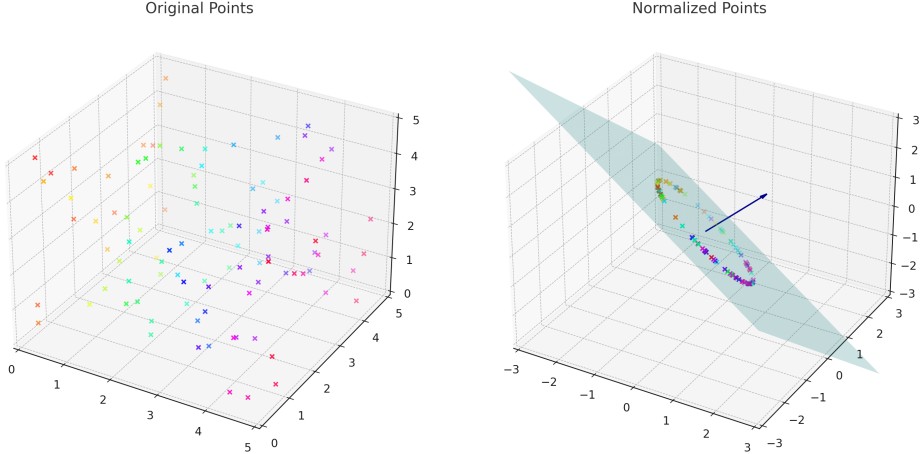

Figure 2: Layer normalization visualized on 3D data. Left: Original feature space (from randomly sampled data), with each data point color-coded according to its position in space. Right: Feature space after layer normalization, note that all data points lie within the plane perpendicular to the $\overrightarrow{\boldsymbol{1}}$ vector.

In modern implementations of the transformer, layer normalization is applied before both the attention and feed-forward module updates within each block, and once more before the final prediction step Xiong et al. (2020). We note that such an arrangement ensures that data within each transformer layer is constrained to the surface of a potentially distinct hyper-sphere. Yet, due to the residual nature of transformers, all intermediate layer representations inhabit the same vector space. As a result, features from different layers project onto a shared hyper-sphere, which we denote as $\mathcal{H}_S$. Interestingly, layer normalization's placement prior to the classification softmax has another consequence. It drives the model to optimize dot-product similarity between certain points within $\mathcal{H}_S$ and word vectors in the embedding matrix $\boldsymbol{W}_E \in \mathbb{R}^{|V| \times d}$, where $|V|$ is the vocabulary size. This optimization indirectly defines the meaning of points in $\mathcal{H}_S$ as a function of their similarity with words represented in $\boldsymbol{W}_E$.

## 2.2 MULTI-HEAD SELF-ATTENTION

To understand how the geometric intuition behind $\mathcal{H}_S$ allows for the interpretability of latent representations within a transformer, we analyze the parameter matrices in the multi-head self-attention module (Vaswani et al., 2017). For a given input sequence $\boldsymbol{X} \in \mathbb{R}^{s \times d}$ of length $s$, the multi-head self-attention mechanism is defined as follows:

$$\text{MultiHead}(\boldsymbol{X}) = \sum_i^h \text{softmax}\left(\frac{\boldsymbol{X}\boldsymbol{W}_{QK}^i\boldsymbol{X}^T}{\sqrt{d}}\right)\boldsymbol{X}\boldsymbol{W}_{VO}^i \tag{5}$$

Where $h$ is the number of self-attention heads while $\boldsymbol{W}_{QK}^i \in \mathbb{R}^{d\times d}$ and $\boldsymbol{W}_{VO}^i \in \mathbb{R}^{d\times d}$ are low-rank virtual matrices obtained by grouping the query, key, value and output projection matrices at each head (Elhage et al., 2021; Dar et al., 2022). A full derivation of Equation 5 from the original formulation by Vaswani et al. (2017) is provided in Appendix B.

### 2.2.1 THE QUERY-KEY MATRIX

For any given head $i$, the query-key matrix $\boldsymbol{W}_{QK}^i$ is commonly interpreted as a bi-linear form $g_i : \mathbb{R}^d \times \mathbb{R}^d \to \mathbb{R}$ that represents the relevance between keys and queries. However, it is also possible to consider $\boldsymbol{W}_{QK}^i$ as a linear transformation that maps inputs to a query representation $\boldsymbol{X}_Q^i = \boldsymbol{X}\boldsymbol{W}_{QK}^i$, similar to that considered in Brody et al. (2023)[1]. With the head's attention score matrix $\boldsymbol{A}^i \in [0,1]^{s\times s}$, for a given sequence length $s$, obtained as:

$$\boldsymbol{A}^i = \text{softmax}\left(\frac{\boldsymbol{X}_Q^i\boldsymbol{X}^T}{\sqrt{d}}\right) \tag{6}$$

This process is illustrated for normalized inputs in the right-most section of Figure 3. Essentially, the role of the $\boldsymbol{W}_{QK}$ matrix and the layer normalization parameters is to find an affine transformation over $\mathcal{H}_S$ such that, when superimposed on itself, brings related terms closer together and keeps unrelated terms apart.

It is important to mention that for $k < d$, the matrix $\boldsymbol{W}_{QK}^i$ cannot be inverted, as it won't have a full rank. This implies, by the rank-nullity theorem, that for each head, there must be a set of $d-k$ query vectors $\mathbb{Q}_{null}^i \subset \mathbb{R}^d$ that map to the zero vector and, as a consequence, attend to all keys equally. Conversely, there must also exist a set of $d-k$ keys $\mathbb{K}_{null}^i \subset \mathbb{R}^d$ that are attended to by all queries equally, with a pre-softmax attention score of zero.

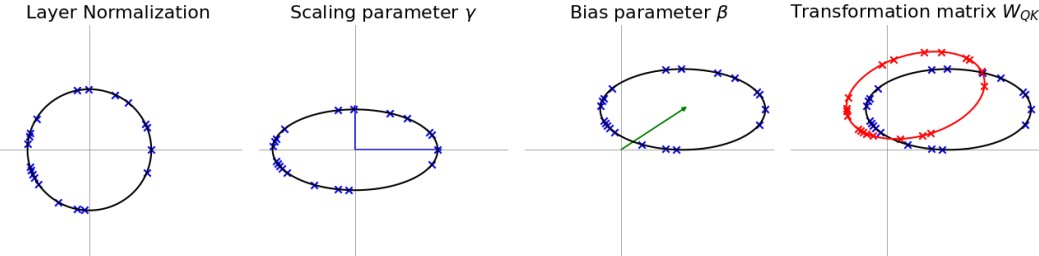

Figure 3: Visualization of the self-attention process for a single head. **Left**: Layer normalization projects the input features on the surface of the hyper-sphere $\mathcal{H}_S$. **Center Left**: A scaling parameter $\gamma$ is commonly applied after normalization; it transforms the hyper-sphere into a hyper-ellipsoid. **Center Right**: A bias term $\boldsymbol{\beta}$ is also applied after normalization; it displaces the hyper-ellipsoid away from the origin. **Right**: The input features are mapped to a query representation (in red) by the matrix $\boldsymbol{W}_{QK}$ and superimposed over their previous representation to obtain the self-attention scores.

### 2.2.2 THE VALUE-OUTPUT MATRIX AND THE RESIDUAL STREAM

To understand the role of the $\boldsymbol{W}_{VO}$ matrix within the transformer, we now consider the update step after multi-head attention at a given layer $l$:

$$\boldsymbol{X}_{l+1} = \boldsymbol{X}_l + \text{MultiHead}(\text{LayerNorm}(\boldsymbol{X})) \tag{7}$$

---

[1]An alternative key representation $\boldsymbol{X}_K^i = \boldsymbol{X}\boldsymbol{W}_{QK}^i{}^T$ can also be considered

Note that by plugging in Equation 5 and Equation 6, the layer update can be re-written as:

$$\boldsymbol{X}_{l+1} = \boldsymbol{X}_l + \sum_i^h \boldsymbol{A}^i \boldsymbol{X}_V^i \tag{8}$$

where

$$\boldsymbol{X}_V^i = \text{LayerNorm}(\boldsymbol{X}_l)\boldsymbol{W}_{VO}^i \tag{9}$$

It can be seen that the multi-head attention mechanism consists of the sum of $h$ individual updates, each one given by one of the attention heads. Within each head, all words in the sequence propose an update $\boldsymbol{X}_V^i$, and these are aggregated according to their attention scores $\boldsymbol{A}^i$. In Equation 9, the matrix $\boldsymbol{W}_{OV}^i$ transforms the normalized inputs in $\mathcal{H}_S$ into a set of updates in the same latent space as the residual stream. Furthermore, we propose that the $\boldsymbol{W}_{VO}^i$ matrix is better understood as a second key-value store (Sukhbaatar et al., 2015; Geva et al., 2021) within the attention layer. To see why, consider its Singular Value Decomposition (SVD) (Millidge & Black, 2022): $\boldsymbol{W}_{VO}^i = \boldsymbol{U}\boldsymbol{\Sigma}\boldsymbol{V}^T$. By substituting in Equation 9, we obtain:

$$\boldsymbol{X}_V^i = (\boldsymbol{Q}_{VO}\boldsymbol{K}_{OV}^i{}^T)\boldsymbol{V}_{OV}^i \tag{10}$$

where

$$\begin{aligned} \boldsymbol{Q}_{VO} &= \text{LayerNorm}(\boldsymbol{X}) \\ \boldsymbol{K}_{OV}^i &= (\boldsymbol{U}\boldsymbol{\Sigma})^T \\ \boldsymbol{V}_{OV}^i &= \boldsymbol{V}^T \end{aligned} \tag{11}$$

The left singular vectors, associated with the columns of $\boldsymbol{U}\boldsymbol{\Sigma} \in \mathbb{R}^{d \times d}$, act as a library of "keys" $\boldsymbol{K}_{OV}^i$ against which the normalized features $\boldsymbol{X}_l \in \mathcal{H}_S$ are compared. While the corresponding right singular vectors, associated with rows in $\boldsymbol{V}^T \in \mathbb{R}^{d \times d}$, act as the output values $\boldsymbol{V}_{OV}^i$ that define the direction in which to update the information in the residual stream for a given key. This interpretation is motivated by the results of Millidge & Black (2022), where it is shown that the right singular vectors $\boldsymbol{V}^T$ of the $\boldsymbol{W}_{VO}$ matrix tend to have interpretable meanings when decoded using $\boldsymbol{W}_E$, with some of the transformer heads consistently representing a single topic in most of their singular vectors. We would also like to mention that, similar to the $\boldsymbol{W}_{QK}$ matrix, the $\boldsymbol{W}_{OV}$ matrix has at least $d - k$ singular values equal to zero. This means that multiple queries $\boldsymbol{Q}_{VO}$ will map to the zero vector and thus won't update the information in the residual stream, allowing the model to skip the update process if necessary.

To conclude, we highlight that the proposed interpretation of attention behaves very similarly to that of the feed-forward module given by Geva et al. (2021), as both calculate relevance scores and aggregate sub-updates for the residual stream. However, the way the scores and updates are calculated is very different. The attention module relies primarily on dynamic context for its scores and values, while the feed-forward module relies on static representations.

## 2.3 THE WORD EMBEDDING MATRIX AND OUTPUT PROBABILITIES

Once all the attention and feed-forward updates have been applied, the output probabilities of the network can be obtained as follows (Xiong et al., 2020):

$$\boldsymbol{P}_Y = \text{softmax}\big(\text{LayerNorm}(\boldsymbol{X}_L)\boldsymbol{W}_E^T\big) \tag{12}$$

Equation 12 can be interpreted as measuring the similarity between the final layer representation $\boldsymbol{X}_L$ when projected to $\mathcal{H}_S$, and each of the embedding vectors in $\boldsymbol{W}_E$. Given that all vectors in the projection have the same norm $\sqrt{d}$, the only relevant factor in deciding the output probability distribution $\boldsymbol{P}_{Y[t,:]} \in [0,1]^{|V|}$, at a given timestep $t$, is the location of its corresponding vector $\boldsymbol{X}_{L[t,:]}$ within $\mathcal{H}_S$. This behavior is very similar to that described by the von Mises-Fisher distribution (Fisher, 1953), as both represent distributions parameterized by a reference vector within a hypersphere. Nonetheless, in the case of transformers, the support of the distribution is defined over a discrete set of points in $\mathbb{R}^d$ instead of the entire surface of $\mathcal{H}_S$, as it is for the von Mises-Fisher.

Table 1: Distance between the normalized embeddings LayerNorm($\boldsymbol{W}_E$) and different transformations of the embedding matrix $W_E$.

| Setting | Mean $\ell_2$ Distance | Mean Cosine Distance |
|---|---|---|
| Original | 23.747 (0.432) | <0.001 (<0.001) |
| Centered | 24.872 (0.432) | 0.150 (0.035) |
| Scaled by $\sqrt{d}$ | 2.413 (1.862) | <0.001 (<0.001) |
| Centered + Scaled by $\sqrt{d}$ | 14.591 (1.469) | 0.150 (0.035) |

In the case where layer normalization includes scaling and bias parameters $\gamma$ and $\boldsymbol{\beta}$, the output probabilities are calculated as follows:

$$\boldsymbol{P}_Y = \text{softmax}\big(\hat{\boldsymbol{X}}_L\boldsymbol{\Gamma}\boldsymbol{W}_E^T + \boldsymbol{\beta}\boldsymbol{W}_E^T\big) \tag{13}$$

where $\hat{\boldsymbol{X}}_L$ is the projection of $\boldsymbol{X}_L$ to $\mathcal{H}_S$ and $\boldsymbol{\Gamma}$ is a diagonal matrix such that $\Gamma_{ii} = \gamma_i$. The effect of $\boldsymbol{\Gamma}$ on the representation is that of transforming $\mathcal{H}_S$ into an ellipsoid (see the center-left section of Figure 3) while $\boldsymbol{\beta}\boldsymbol{W}_E^T$ acts as a bias that assigns higher probability to certain tokens independent of the input.

In both cases (with and without bias and scale parameters), the proposed interpretation aligns with that of iterative refinement within transformers (Jastrzebski et al., 2017; nostalgebraist, 2020; Elhage et al., 2021; Geva et al., 2022; Belrose et al., 2023), given that intermediate representations $\boldsymbol{X}_l$ can always be converted into output probabilities using Equation 12.

## 3  EXPERIMENTS

This section presents our experimental results. All experiments were done on a RTX 4090 GPU using pre-trained weights from the 124M parameter version of GPT-2 (Radford et al., 2019; Karpathy, 2023) [2].

### 3.1  IMPACT OF LAYER NORMALIZATION ON THE WORD EMBEDDINGS

To measure the impact of layer normalization on the position of the embedding vectors $\boldsymbol{w}_e \in \boldsymbol{W}_E$, we calculated both the $\ell_2$ and cosine distances between the layer-normalized weights and the following settings:

- Original: The original word embeddings without any modification
- Centered: Original + centering around the mean $\text{E}[\boldsymbol{w}_e]$
- Scaled: Original divided by the average vector norm $\text{E}[||\boldsymbol{w}_e||]$ and multiplied by $\sqrt{d}$
- Centered + Scaled: Original + centering + scaling

The results in Table 1 show that the mean cosine distance between the original word embeddings and the embeddings after normalization is close to zero, meaning that projection onto $\mathcal{H}_S$ does not modify the orientation of the embedding vectors. The results also confirm this when centering is applied, as the cosine distance increases significantly when the original vectors are displaced from the origin and towards the mean. On the other hand, it can be seen that the $\ell_2$ distance is high for all settings except for when scaling is applied without centering. Given an average norm of $\text{E}[||\boldsymbol{w}_e||] = 3.959$ and for $\sqrt{d} = 27.713$ we can conclude that the original word embeddings lie between the origin and $\mathcal{H}_S$ rather than on its surface, with different embeddings having different norms.

We hypothesize that variance in the norm of embedding vectors ($\text{SD}(||\boldsymbol{w}_e||) = 0.434$) is likely to be a result of the use of the word embedding matrix as a classification layer (see Equation 13). To verify whether this is the case, we select the top and bottom 5 embedding vectors based on the three following criteria:

---

[2]Code to replicate all experiments will be made available upon acceptance

Table 2: Top 5 and Bottom 5 tokens from the word embedding matrix.

| Position | Norm | Scaled Norm | Norm + Bias | Scaled Norm + Bias |
|---|---|---|---|---|
| Top 1 | SPONSORED | \xa9\xb6\xe6 | , | the |
| Top 2 | \x96\x9a | tremend | the | , |
| Top 3 | soDeliveryDate | \x96\x9a | . | and |
| Top 4 | enegger | senal | and | a |
| Top 5 | Reviewer | millenn | - | in |
| Bottom 5 | for | - | \xc0 | \x07 |
| Bottom 4 | an | ( | \x07 | \x0f |
| Bottom 3 | on | "\n" | \x10 | oreAndOnline |
| Bottom 2 | in | , | \x11 | \x06 |
| Bottom 1 | at | . | \xfe | \xc1 |

- Norm: The norm of the original embedding vector $\boldsymbol{w}_E$
- Scaled Norm: The norm of the embedding vector when scaled by the Layer Norm parameter $\boldsymbol{\Gamma}$
- Norm + Bias: The norm of the original embedding vector plus the bias scores obtained from $\boldsymbol{\beta W}_E^T$
- Scaled Norm + Bias: The sum between the Scaled Norm and the bias scores.

The sorted tokens in Table 2 show that considering only the norm of the embeddings is not enough, as tokens that are not commonly used (like 'SPONSORED' and 'soDeliveryDate') have the highest norms, while common words like 'for', 'an', 'on' and 'in' have the lowest norm. After considering the scaling parameter $\boldsymbol{\Gamma}$, we observe that punctuation signs like the newline character or the comma ',' have the lowest norm, and that there is no clear pattern on the top tokens. After considering bias, we see that the distribution of top tokens clearly shifts, with punctuation symbols and common words now at the top and uncommon bytes at the bottom. Finally, note that when both scale and bias are considered, the top tokens are consistent with some of the most common words in the English language: 'the', 'and', 'a' and 'in' with the only exception being the comma character, which is also very common in natural language, while the bottom tokens are related to uncommon bytes and an anomalous token.

### 3.2 PROBING ATTENTION HEADS WITH NORMALIZED REPRESENTATIONS OF COMMON NOUNS

Next, we probe the attention heads at layers 0, 5 and 11 of the GPT-2 model using as inputs the 100 most common nouns taken from the Corpus of Contemporary American English (COCA) (Davies, 2010). First, we transform the embedding matrix $\boldsymbol{W}_E$ according to the normalization parameters specific to each layer (see Figure 3) and then multiply the normalized embeddings $\hat{\boldsymbol{W}}_E$ by either $\boldsymbol{W}_{QK}$ or $\boldsymbol{W}_{VO}$. To decode the output from $\boldsymbol{W}_{QK}$, we retrieve the top-k closest embedding vectors from $\hat{\boldsymbol{W}}_E$ based on dot product similarity. For $\boldsymbol{W}_{VO}$, we add the head-specific and layer-specific output biases (see Equation S.8) to obtain the "update vectors". These update vectors are then added to the original embeddings from $\boldsymbol{W}_E$ and transformed according to the normalization parameters from the last layer; then, we retrieve the top-k closest embeddings from the original $\boldsymbol{W}_E$ embedding matrix based on dot product similarity.

#### 3.2.1 QUERY-KEY TRANSFORMATIONS

In Table D. 1, we present partial results for the query-key transformations at layer 0, given the query inputs 'time', 'life' and 'world'. We note that some of the heads preserve the meaning of the query, as is the case for heads 1, 5 and 10, possibly looking for repetition, while others look for keys that precede it. Such precedence heads might help to disambiguate the meaning of the words, with examples like: 'Showtime' vs. 'spacetime', 'battery life' vs. 'wildlife' and 'underworld' vs. 'Westworld'. Other heads appear to be looking for contextual associations, as is the case for head 2, which seems to relate 'world' with dates and concepts from the First and Second World wars. When looking at deeper layers (as shown in Table D. 2 & D. 3), we were not able to identify any meaningful patterns on the query transformations, suggesting that these layers might look for more complex patterns that cannot be measured by probing.

### 3.2.2 KEY-VALUE TRANSFORMATIONS

In Table D. 4, we present partial results for the key-value transformations using the same three sample inputs. For most heads at layer 0, the meaning of the input key is kept as is. However, when the sum of all the heads is considered, we see a slight shift in the meaning of the words. For heads at layer 5 (shown in Table D. 5), we see that although most of the heads preserve the meaning of the input keys 'life' and 'world' (and around half of the heads for the input 'time'), the sum of all heads does change the word meaning dramatically, and without a clear output pattern. As our experiment is limited to testing a single input key at a time, it might be possible that updates in this layer rely more heavily on the contextual composition between multiple keys, which we did not capture. Finally, in the last layer (Table D. 6), we see that most individual heads map to seemingly arbitrary values, with only a few preserving the meaning of the input key. However, when the sum of the heads is considered, the layer preserves the meaning of the input keys. To test the hypothesis that meaning-preserving heads dominated the layer update, we measured the norm of the output values for each head (before adding the layer-specific bias $\beta_O$). We found that, in most cases, these heads do not have higher norms. Instead, heads promoting common tokens like 'the', ',' and 'and' had the highest norms. These results suggest that some heads at the last layer work together to preserve the meaning of the input keys and mitigate the network's bias towards common tokens.

### 3.3 PROBING THE SINGULAR VECTORS OF THE VIRTUAL ATTENTION MATRICES

#### 3.3.1 SINGULAR VECTORS OF THE KEY-VALUE MATRIX

To verify whether the key-value interpretation of $W_{VO}$ matrix proposed in subsubsection 2.2.2 is correct, we probe each of its singular vectors (as proposed in Millidge & Black (2022)). For the left singular vectors $U$ (scaled by $\Sigma$), we use the normalized embeddings $\hat{W}_E$ as a probe, while for the right singular vectors $V^T$, we use the original embeddings $W_E$. Given that all singular values are constrained to be positive, we get two possible singular vector pairs corresponding to each singular value: $(u, v)$ and $(-u, -v)$. For ease of analysis, we choose the signed pair with its $v$ component closest to any of the embeddings $w_e \in W_E$, using the dot product similarity.

We did not observe any interpretable pattern for the attention heads at layer 0 and found only one interpretable head at layer 5 (head 10), which referred to terms in politics and chemistry. However, we found that most heads in layer 11 were interpretable (except for heads 5, 7 and 9) and present the results for all heads in Appendix E. An illustrative case of these patterns is head 3, where most of its singular vector mappings are related to jobs or industries. For example, 'Dairy' maps to 'USDA' (the United States Department of Agriculture), 'engine' to 'drivers', 'trading' to 'Sales' and so on. Similar patterns were present in other heads, listed as follows:

- **Head 0:** Formatting and punctuation symbols (end of text, new line, brackets and parenthesis)
- **Head 1:** Gender words
- **Head 2:** Proper Nouns (Places)
- **Head 3:** Jobs / Industries
- **Head 4:** Letters and Numbers

- **Head 6:** Suffixes and Prefixes related to the ending and beginning of words
- **Head 8:** Punctuation symbols
- **Head 10:** Proper Nouns (First and Last names)
- **Head 11:** The identity function (input similar to the output)

We found that these patterns were consistent with those obtained in the key → value results from Table D. 6, implying that the subject-specific behavior of the singular vectors is reflected in the input-output transformations of the attention heads. These results complement previous work from Millidge & Black (2022), in which only the right singular vectors $V^T$ were considered.

#### 3.3.2 SINGULAR VECTORS OF THE QUERY-KEY MATRIX

In additional experiments on the SVD of the $W_{QK}$ matrix, we found that some singular vector pairs had clear associations. For example, in head 0 of layer 0, we found some associations related to programming languages ('self, class, =, import' → 'Python') and digital cameras ('Video, 264, minutes' → 'Nikon, lineup, shot, camera') but we could not identify any specialization for the heads

in this layer. Surprisingly, we did find that heads at the last layer had identifiable patterns on their left singular vectors (associated with the queries) consistent with those listed for the $\boldsymbol{W}_{VO}$ matrix (punctuation for head 0, gender for head 1, and so on), but no clear patterns were identified for the right singular vectors.

### 3.4 VISUALIZING ITERATIVE REFINEMENT

Finally, we visualize how the information in the residual stream is updated (i.e. the iterative refinement process) leveraging dimensionality reduction techniques, as shown in Figure 4. For this, we chose the test sentence 'To kill two birds with one stone', as the predictability of its last token, 'stone', given the previous context was high (correctly predicted by the model) and none of the words in the sentence repeated. To project the high dimensional embeddings into 3D space, we used UMAP (McInnes et al., 2018), with Laplacian Eigenmap initialization (Belkin & Niyogi, 2001; Kobak & Linderman, 2021), and we fit the transform using the first 10,000 embedding vectors from $\boldsymbol{W}_E$ to accurately reflect proximity in the original embedding space. We show the original embedding tokens as reference (in blue) and plot the trajectory of the second-to-last token, 'one', as we process the entire sequence (with added positional embeddings) throughout the network. For each layer, we transform the latent representations in the residual stream using the normalization parameters from the final output layer before projecting with UMAP. It can be

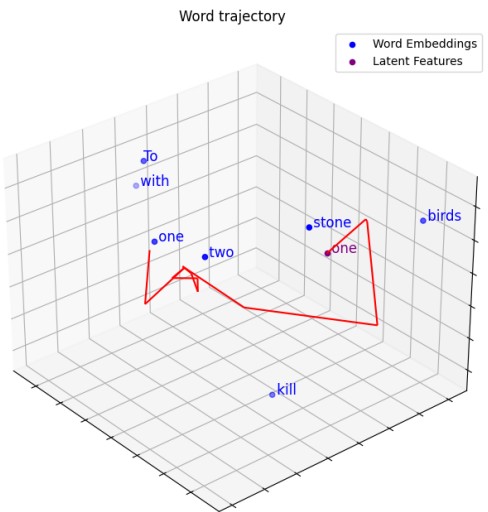

Figure 4: UMAP 3D projection of the phrase 'To kill two birds with one stone'.

seen that the representation of the second-to-last token shifts from its original meaning ('one') towards the meaning of the next token ('stone'). Although the figure also shows the magnitude and direction of each update in the trajectory, it is important to mention that these quantities might have been modified due to the dimensionality reduction process.

## 4 CONCLUSION

We have presented a new interpretation of transformer models based on the geometric intuition behind each of its components. First, we showed how layer normalization can be better understood as a projection of latent features in $\mathbb{R}^d$ to a $(d-1)$-dimensional hyper-sphere and provide experimental evidence that the word embeddings learned by GPT-2 are distributed toward different directions of the hyper-sphere, we also demonstrate that the parameters of the final normalization layer are crucial in obtaining high-scoring tokens consistent with high-frequency tokens in the English language. Second, we discussed the role of the $\boldsymbol{W}_{QK}$ and $\boldsymbol{W}_{VO}$ matrices as transformations related to the hyper-sphere, with $\boldsymbol{W}_{QK}$ as an affine transformation that overlaps queries and keys, and $\boldsymbol{W}_{VO}$ as a key-value map between the hyper-sphere and the original embedding space. These intuitions were experimentally tested with probing experiments, showing promising results in understanding the role of query-key attention in earlier layers and extending the results from Millidge & Black (2022) on the subject-specific nature of the $\boldsymbol{W}_{VO}$ matrix in attention heads at deeper layers. Finally, we integrated these ideas and the impact of each component on the residual stream to provide visual evidence of how the iterative refinement process works within transformers, illustrating the journey that occurs from a previous token to the next.

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

## A GEOMETRIC TRANSFORMATIONS WITHIN LAYER NORMALIZATION

### A.1 MEAN ALONG DIMENSIONS AS VECTOR PROJECTION

If we consider the unit-norm vector $\frac{1}{\sqrt{d}}\overrightarrow{\mathbf{1}}$, it can be shown that the mean along dimensions $\boldsymbol{\mu}$ is the projection of $\boldsymbol{x}$ onto $\frac{1}{\sqrt{d}}\overrightarrow{\mathbf{1}}$:

$$
\begin{aligned}
\operatorname{proj}(\boldsymbol{x}, \frac{1}{\sqrt{d}}\overrightarrow{\mathbf{1}}) &= \frac{1}{||\frac{1}{\sqrt{d}}\overrightarrow{\mathbf{1}}||_2}\left(\boldsymbol{x} \cdot \frac{1}{\sqrt{d}}\overrightarrow{\mathbf{1}}\right)\frac{1}{\sqrt{d}}\overrightarrow{\mathbf{1}} \\
&= \left(\boldsymbol{x} \cdot \frac{1}{\sqrt{d}}\overrightarrow{\mathbf{1}}\right)\frac{1}{\sqrt{d}}\overrightarrow{\mathbf{1}} \\
&= \left(\frac{\boldsymbol{x} \cdot \overrightarrow{\mathbf{1}}}{\sqrt{d}}\right)\frac{1}{\sqrt{d}}\overrightarrow{\mathbf{1}} \\
&= \left(\frac{\boldsymbol{x} \cdot \overrightarrow{\mathbf{1}}}{d}\right)\overrightarrow{\mathbf{1}} \\
&= \left(\frac{1}{d}\sum_i^d x_i\right)\overrightarrow{\mathbf{1}} \\
&= \mu\overrightarrow{\mathbf{1}} \\
&= \boldsymbol{\mu}
\end{aligned}
\tag{S.1}
$$

For $\boldsymbol{\mu} = [\mu, \mu, \ldots, \mu] \in \mathbb{R}^d$.

### A.2 STANDARD DEVIATIONS AS NORM SCALING

Brody et al. (2023) show that the division by $\sigma$ acts as a scaling factor that modifies the norm of $\boldsymbol{x} - \mu$ to be $\sqrt{d}$:

$$
\begin{aligned}
\sigma &= \sqrt{\frac{1}{d}\sum_i^d (x_i - \mu)^2} \\
&= \frac{1}{\sqrt{d}}\sqrt{\sum_i^d (x_i - \mu)^2} \\
&= \frac{1}{\sqrt{d}}||\boldsymbol{x} - \boldsymbol{\mu}||_2
\end{aligned}
\tag{S.2}
$$

# B    ADDITIVE FORMULATION OF MULTI-HEAD SELF-ATTENTION

Vaswani et al. (2017) define the self-attention mechanism as follows:

$$\text{SelfAttention}(\boldsymbol{X}, \boldsymbol{W}_Q, \boldsymbol{W}_K, \boldsymbol{W}_V) = \text{softmax}\left(\frac{\boldsymbol{Q}\boldsymbol{K}^T}{\sqrt{d}}\right)\boldsymbol{V} \tag{S.3}$$

where

$$\begin{aligned} \boldsymbol{Q} &= \boldsymbol{X}\boldsymbol{W}_Q \\ \boldsymbol{K} &= \boldsymbol{X}\boldsymbol{W}_K \\ \boldsymbol{V} &= \boldsymbol{X}\boldsymbol{W}_V \end{aligned} \tag{S.4}$$

Such that $\boldsymbol{W}_Q \in \mathbb{R}^{d \times k}$, $\boldsymbol{W}_K \in \mathbb{R}^{d \times k}$ and $\boldsymbol{W}_V \in \mathbb{R}^{d \times v}$ are projection matrices from the original model dimension $d$ to intermediate dimension $k$ and value dimension $v$, respectively. For multi-head attention, multiple projection matrices $\boldsymbol{W}_Q^i$, $\boldsymbol{W}_K^i$, $\boldsymbol{W}_V^i$ are considered, one for each head $i \in [1, \ldots, h]$ (with $h$ being the number of heads). In this case, the value dimension $v$ is commonly set equal to $k$ and an extra projection matrix $\boldsymbol{W}_O \in \mathbb{R}^{hk \times d}$ is introduced to combine information from all heads as follows (Vaswani et al., 2017):

$$\begin{aligned} \text{MultiHead}(\boldsymbol{X}) &= \text{Concat}([\text{head}_1, \ldots, \text{head}_h])\boldsymbol{W}_O \\ \text{where head}_i &= \text{SelfAttention}(\boldsymbol{X}, \boldsymbol{W}_Q^i, \boldsymbol{W}_K^i, \boldsymbol{W}_V^i) \end{aligned} \tag{S.5}$$

Given that the concatenation happens along the row dimension of each head, it is possible to re-write multi-head self-attention as follows:

$$\begin{aligned} \text{MultiHead}(\boldsymbol{X}) &= \sum_i^h \text{SelfAttention}(\boldsymbol{X}, \boldsymbol{W}_Q^i, \boldsymbol{W}_K^i, \boldsymbol{W}_V^i)\boldsymbol{W}_O^i \\ \text{where } \boldsymbol{W}_O &= \text{Concat}[\boldsymbol{W}_O^1, \ldots, \boldsymbol{W}_O^h] \end{aligned} \tag{S.6}$$

Such that each $\boldsymbol{W}_O^i \in \mathbb{R}^{k \times d}$ denotes an element of the partition of matrix $\boldsymbol{W}_O$ alongside the row dimension. Combining Equation S.4 and Equation S.6 we obtain a single formula for multi-head self-attention:

$$\begin{aligned} \text{MultiHead}(\boldsymbol{X}) &= \sum_i^h \text{softmax}\left(\frac{\boldsymbol{X}\boldsymbol{W}_Q^i {\boldsymbol{W}_K^i}^T \boldsymbol{X}^T}{\sqrt{d}}\right)\boldsymbol{X}\boldsymbol{W}_V^i\boldsymbol{W}_O^i \\ &= \sum_i^h \text{softmax}\left(\frac{\boldsymbol{X}\boldsymbol{W}_{QK}^i\boldsymbol{X}^T}{\sqrt{d}}\right)\boldsymbol{X}\boldsymbol{W}_{VO}^i \end{aligned}$$

## C  Impact of Bias Terms in Multi-Head Self-Attention

### C.1  Bias in the Query and Key Projections

In case the projection given by Equation S.4 contains bias terms $\boldsymbol{\beta}_Q, \boldsymbol{\beta}_K \in \mathbb{R}^k$, the attention score matrix from Equation 6 is calculated as follows:

$$\boldsymbol{A}^i = \text{softmax}\left(\frac{\boldsymbol{X}_Q^i \boldsymbol{X}^T + \boldsymbol{X} \boldsymbol{W}_Q^i \boldsymbol{\beta}_K^T + \boldsymbol{\beta}_Q \boldsymbol{W}_k^{iT} \boldsymbol{X}^T + \boldsymbol{\beta}_Q \boldsymbol{\beta}_K^T}{\sqrt{d}}\right) \tag{S.7}$$

In the bias formulation, three new terms are introduced. First, $\boldsymbol{W}_Q^i \boldsymbol{\beta}_K^T \in \mathbb{R}^d$, which can be thought of as a reference vector for queries, such that queries similar to it get higher attention scores. Given that the same "bias score" will be broadcasted along all the different keys of the same query, the network will ignore this term due to the shift-invariance of the softmax function. More interesting is the second term $\boldsymbol{\beta}_Q \boldsymbol{W}_K^{iT} \in \mathbb{R}^d$, which acts as a reference for keys. Given that its bias score is broadcasted along queries, it will result in higher attention scores (in all queries) for keys similar to the reference. Finally, the term $\boldsymbol{\beta}_Q \boldsymbol{\beta}_K^T \in \mathbb{R}$ acts as a global bias and, similar to $\boldsymbol{W}_Q^i \boldsymbol{\beta}_K^T$, will be ignored by the network.

### C.2  Bias in the Value and Output Projections

If the value projection in Equation S.4 contains a bias term $\boldsymbol{\beta}_V \in \mathbb{R}^k$, and the output projection in Equation S.5 contains a bias term $\boldsymbol{\beta}_O \in \mathbb{R}^d$. The layer update in Equation 8 can be re-written as follows:

$$\boldsymbol{X}_{l+1} = \boldsymbol{X}_l + \boldsymbol{\beta}_O + \sum_i^h \boldsymbol{A}^i \boldsymbol{X}_{value}^i + \boldsymbol{\beta}_V \boldsymbol{W}_O^{iT} \tag{S.8}$$

Here, the term $\boldsymbol{\beta}_V \boldsymbol{W}_O^{iT} \in \mathbb{R}^d$ is a bias on the update direction of head $i$, while $\boldsymbol{\beta}_O \in \mathbb{R}^d$ acts as a bias on the entire layer's update.

# D    ATTENTION HEAD TRANSFORMATIONS FOR LAYERS 5 AND 11

## D.1    QUERY-KEY TRANSFORMATIONS

Table D. 1: Transformation of Queries Across Attention Heads at Layer 0

| | Query → Keys | | |
|---|---|---|---|
| **Head** | time | life | world |
| 0 | Level, [?], offenders | battery, Battery, Battery | legraph, Vers, Malf |
| 1 | time, time, Time | Life, life, life | World, world, world |
| 2 | cinematic, Priest, priest | Notre, fetal, abortion | 1914, Churchill, 1916 |
| 3 | space, lunch, mid | augh, ertain, ough | under, Nether, Fort |
| 4 | soft, heavy, tool | Middle, Hans, Middle | ether, Unt, Know |
| 5 | time, time, Time | life, Life, Life | world, World, world |
| 6 | Rated, chirop, u | Fukushima, chirop, ulic | ipt, u, Meta |
| 7 | Show, bed, Movie | pro, wild, Wild | Disc, West, West |
| 8 | java, framework, watch | shark, sharks, Wild | edit, ”$:/, movie |
| 9 | stones, pal, cards | Trojan, malware, Wi | Rogers, COUNTY, Rd |
| 10 | time, time, Time | life, life, Life | world, world, World |
| 11 | Wine, a, food | PHI, everal, Span | agus, true, ‘,’ |

Table D. 2: Transformation of Queries Across Attention Heads at Layer 5

| | Query → Keys | | |
|---|---|---|---|
| **Head** | time | life | world |
| 0 | depend, annot, reason | so, inf, char | Lab, dev, Dev |
| 1 | they, themselves, Vers | they, Im, depend | come, once, haven |
| 2 | Nepal, ”:[”, —” | ‘. . . .’, ‘. . . ’, Home | posted, Logged, ideologically |
| 3 | appeared, actually, had | posted, axle, .avascript | aryl, Ala, GA |
| 4 | attract, CP, contained | misconception, (?, trophy | separatists, activists, extremists |
| 5 | Plum, rice, Vers | Sniper, too, hides | Prim, Bright, am |
| 6 | en, annually, – | following, Generator, Library | §§, tournaments, StarCraft |
| 7 | Wis, def, individual | y, ier, od | Af, Gh, agle |
| 8 | condition, intensive, inf | prol, operation, splend | Ard, marketplace, dev |
| 9 | post, market, destinations | She, steal, etc | strategy, pd, budget |
| 10 | jugg, continuously, Center | essim, enter, tast | exploration, jugg, PLAY |
| 11 | straight, interview, fucking | –, Eva, related | spotlight, television, TV |

Table D. 3: Transformation of Queries Across Attention Heads at Layer 11

| | Query → Keys | | |
|---|---|---|---|
| **Head** | time | life | world |
| 0 | UNCLASSIFIED, opausal, ster | opausal, backstage, piece | routine, cat, ocular |
| 1 | assion, upp, pir | pir, Virgin, appa | Frontier, theater, onies |
| 2 | heid, GI, rict | heid, apy, brance | region, urgy, encyclopedia |
| 3 | opic, href, Hitchcock | susceptibility, space, opic | league, space, opic |
| 4 | gy, lots, whatever | his, whichever, gy | whichever, whatever, underworld |
| 5 | olesterol, tx, erc | olesterol, avy, iana | wealth, Digest, Market |
| 6 | ones, volatile, RIS | volatile, olesterol, idency | wealth, useum, theatre |
| 7 | whichever, ivalent, lower | mortal, whichever, living | -$, complex, world |
| 8 | ove, HTTP, metaphysical | spiritual, metaphysical, bio | Endless, metaphysical, Marvel |
| 9 | Productions, actic, fare | stuff, ience, Productions | entertainment, stuff, World |
| 10 | -, code, ing | -, core, ola | core, Labs, ourse |
| 11 | emb, ivan, Union | Tour, etc, iona | pires, si, Tour |

## D.2  KEY-VALUE TRANSFORMATIONS

Table D. 4: Transformation of Keys Across Attention Heads at Layer 0

| Head | Key → Values | | |
|---|---|---|---|
| | time | life | world |
| 0 | time, Time, time | life, choice, senal | world, World, worlds |
| 1 | time, TIME, time | life, lihood, life | world, Goes, ship |
| 2 | time, [?], Minutes | life, Life, life | world, world, World |
| 3 | time, Time, theless | life, Life, life | world, World, worlds |
| 4 | time, time, Time | life, Life, Life | world, World, world |
| 5 | time, Time, Time | life, Life, Life | world, World, worlds |
| 6 | time, time, Time | life, life, Life | world, world, Feather |
| 7 | time, eless, times | life, Experience, Life | world, World, Abyss |
| 8 | time, iversary, melodies | life, challeng, conservancy | world, worlds, droid |
| 9 | time, time, recall | [?], local, Main | [?], world, local |
| 10 | equivalents, igation, planes | life, ento, planner | world, ento, Tanzania |
| 11 | time, Time, Time | life, Life, +++ | world, World, Trials |
| Sum | time, etime, watch | Indigo, life, crew | world, Unleashed, World |

Table D. 5: Transformation of Keys Across Attention Heads at Layer 5

| Head | Key → Values | | |
|---|---|---|---|
| | time | life | world |
| 0 | BuyableInstoreAndOnline, [?], time | life, advertising, Life | world, opathy, qus |
| 1 | MON, Sophia, time | mallow, cause, unn | world, Cav, fect |
| 2 | time, qualified, understatement | life, life, Life | world, World, auri |
| 3 | )?, ¿), ?' | \] =>, life, \\" > | world, \] =>, %" |
| 4 | time, TIME, Sabha | Izan, eworld, ieu | world, izons, orld |
| 5 | destro, time, rall | life, Life, agre | world, toget, enthusi |
| 6 | time, time, TIME | life, Life, life | world, World, WORLD |
| 7 | time, corrid, patch | life, Life, life | world, mathemat, redes |
| 8 | NetMessage, [?], ibu | venge, idth, aten | ULTS, Magikarp, [?] |
| 9 | [?], [?], amina | raviolet, los, SPONSORED | Kraft, quickShipAvailable, Berks |
| 10 | time, contrace, Symphony | life, Life, life | world, World, worlds |
| 11 | otle, ide, Ide | framing, plot, plots | ittee, rf, pawn |
| Sum | externalActionCode, ]), issance | ahon, awa, ]" | Magikarp, Hig, ETHOD |

Table D. 6: Transformation of Keys Across Attention Heads at Block 11

| Head | Key → Values | | |
|---|---|---|---|
| | time | life | world |
| 0 | \n, ", " | {, ¿, "# | [, [* ,[ |
| 1 | player, party, Party | youth, House, Youth | party, Trump, party |
| 2 | Lisp, Ö, ¨ | [?], Quincy, Yemen | Scotland, Osborne, Scotland |
| 3 | Weather, cinem, weather | life, euth, Life | world, Worlds, geop |
| 4 | b, k, 2 | inav, d, 4 | i, V, Rivals |
| 5 | Part, Show, part | Well, Well, saw | sees, works, View |
| 6 | Sub, AM, BR | West, West, East | Sub, Under, ob |
| 7 | Journal, Air, Online | home, Home, house | home, Home, internet |
| 8 | ',', the, and | ',', the, and | the, ',', and |
| 9 | interaction, impression, experience | encounter, belief, encounters | reservations, Illusion, illusions |
| 10 | time, TIME, Time | life, life, LIFE | world, world, worlds |
| 11 | time, time, Time | life, LIFE, life | world, oy, door |
| Sum | time, Time, time | life, Life, Life | world, Worlds, worlds |

# E $W_{VO}$ SVD PER HEAD FOR LAYER 11

Table E. 7: Left and Right Singular Vectors at Layer 11 - Head 0

| Rank | Top-3 Left Words | Top-3 Right Words |
|---|---|---|
| 0 | shenan, cryst, encount | DragonMagazine, ertodd, soDelivery-Date |
| 1 | another, Iv, sil | trave, BuyableInstoreAndOnline, con-vol |
| 2 | Sebastian, Luke, humankind | quickShipAvailable, EStream, MpServer |
| 3 | rans, thereby, hem | BuyableInstoreAndOnline, acknow, Buyable |
| 4 | sectional, [+], Winged | ThumbnailImage, \ufffd\ufffd\u58eb, Orderable |
| 5 | abl, isc, Ah | etheless, olson, llah |
| 6 | <\|endoftext\|>, Advertisements, cest | <\|endoftext\|>, Advertisements, kin-dred |
| 7 | ococ, ilan, guest | pard, MBA, uid |
| 8 | ]., ],, ]; | [, [*, [ |
| 9 | \n\n, ),, cakes | \n\n, Quote, Quote |
| 10 | snaps, Bills, Texans | lineback, Chargers, Packers |
| 11 | )..., ...), )." | (\u00a3, (, (?, |
| 12 | pen, cle, Orioles | .'"', [, .") |
| 13 | ](, drm, Updated | \n\n, [/, [/ |
| 14 | RBI, Field, Triple | RHP, RBI, Negro |
| 15 | pod, illus, Maple | ipeg, aboriginal, "\u2026 |
| 16 | am, 'm, hearted | SPONSORED, Newsletter, .... |
| 17 | Document, whit, Scott | SPONSORED, tsky, Ras |
| 18 | gen, idd, anned | Ukrain, prin, rul |
| 19 | Ryder, icz, abet | istries, plet, Gad |

Table E. 8: Left and Right Singular Vectors at Layer 11 - Head 1

| Rank | Top-3 Left Words | Top-3 Right Words |
|---|---|---|
| 0 | Customers, However, Customer | \u899a\u9192, natureconservancy, racuse |
| 1 | mint, Anne, Marie | hers, actress, Denise |
| 2 | ook, Child, ooks | parents, Parents, Children |
| 3 | gow, abad, BEL | boy, student, Guy |
| 4 | eries, girl, girls | Girl, girl, Queen |
| 5 | Marie, Sue, Patricia | Woman, woman, woman |
| 6 | Him, les, LCS | Person, Persons, Person |
| 7 | ndra, Joint, rity | Her, Her, femin |
| 8 | Coach, recapt, Players | Players, Coach, coaches |
| 9 | istries, WAYS, INAL | god, Allaah, God |
| 10 | Ens, offspring, statute | male, males, Woman |
| 11 | Junction, hole, Abdullah | girl, daddy, Neighbor |
| 12 | HR, ig, akings | Major, Major, minors |
| 13 | reunion, Madison, mes | boys, males, Girls |
| 14 | asting, uba, ynt | mom, moms, Jim |
| 15 | ately, ynam, OUS | doctoral, apprentice, Child |
| 16 | ifier, Come, Weekly | class, owners, Class |
| 17 | Confederation, ATE, ingredient | Students, Students, Ms |
| 18 | athon, jen, candidates | Candidate, candidate, traveler |
| 19 | Pres, ently, Secure | character, Characters, Character |

Table E. 9: Left and Right Singular Vectors at Layer 11 - Head 2

| Rank | Top-3 Left Words | Top-3 Right Words |
|---|---|---|
| 0 | orpor, rul, Bolivia | Adelaide, Edmonton, Calgary |
| 1 | ball, ERY, hem | Filipino, Ultron, ANC |
| 2 | \u30f3\u30b8, else, Lib | Ruby, Scarborough, Erit |
| 3 | verb, Lamar, Ankara | Detroit, Detroit, Wenger |
| 4 | iana, amacare, edia | Zoro, Shelby, Tehran |
| 5 | Gw, otle, Rangers | \u00ed, Jinn, Texans |
| 6 | ration, Rim, ially | Yang, McCain, Yang |
| 7 | detector, OTOS, Petersen | Chilean, Pharaoh, ffen |
| 8 | ald, benefit, ahon | Petersburg, Henderson, Kessler |
| 9 | scope, whe, verse | acio, Mits, Jacobs |
| 10 | Gators, Laden, SEAL | Malfoy, Swanson, Romney |
| 11 | Lilly, \u00e9t, lla | Greenwood, Collins, Byrne |
| 12 | ister, ority, isters | Niagara, Maharashtra, soDeliveryDate |
| 13 | Paulo, nesota, Clayton | Loki, \u011f, Finnish |
| 14 | creen, Cron, Base | Pike, Krishna, Satoshi |
| 15 | lake, SP, seeing | Alberta, Arlington, McKin |
| 16 | Bowie, ystem, rey | Bowie, Murray, Utah |
| 17 | head, ding, ressed | Bulgar, Warcraft, Crimean |
| 18 | Venom, elman, lyn | SJ, Brit, Gordon |
| 19 | wright, ansas, arta | NXT, Metroid, Aether |

Table E. 10: Left and Right Singular Vectors at Layer 11 - Head 3

| Rank | Left Words | Right Words |
|------|-----------|-------------|
| 0 | suburbs, restaur, \ufffd | DragonMagazine, BuyableInstoreAndOnline, \ufffd\u9192 |
| 1 | Dairy, farm, Veget | USDA, Dairy, cows |
| 2 | engine, drivers, Motor | Drivers, drivers, driver |
| 3 | trading, trade, shoppers | Sales, retailers, shoppers |
| 4 | instrument, musical, guitar | Billboard, halftime, Grammy |
| 5 | sail, boat, sailing | sail, sailing, autical |
| 6 | teachers, teacher, school | teachers, uberty, curric |
| 7 | baker, kindergarten, bakery | baker, SERV, kindergarten |
| 8 | apparel, prison, recruiting | Sail, Prison, jail |
| 9 | shelter, indoors, shelters | shelters, shelter, Radiant |
| 10 | tribe, fish, fish | dred, whales, fisheries |
| 11 | workers, jobs, job | workers, worker, subcontract |
| 12 | Derrick, tribe, Tribal | Seg, forest, Derrick |
| 13 | chess, Chess, seating | Chess, chess, Sheldon |
| 14 | Soy, Satellite, astronauts | Soy, Satellite, transmissions |
| 15 | Anim, visa, Imm | exhib, Anim, Imm |
| 16 | medicine, diagnose, doctors | Doctors, hospital, doctor |
| 17 | boxing, trainer, spar | boxing, spar, UFC |
| 18 | gun, firearm, Sheriff | ITV, Decoder, Geral |
| 19 | gambling, tournaments, tournament | gambling, Gaming, tournaments |

Table E. 11: Left and Right Singular Vectors at Layer 11 - Head 4

| Rank | Top-3 Left Words | Top-3 Right Words |
|---|---|---|
| 0 | them, their, him | cloneembedreportprint, \u899a\u9192, \u30b5\u30fc\u30c6\u30a3 |
| 1 | iator, ive, ibur | natureconservancy, Canaver, \u25fc |
| 2 | if, born, forces | the, ., , |
| 3 | ually, ,., therein | Buyable, misunder, lehem |
| 4 | irk, struct, actly | 1, 2, 9 |
| 5 | uku, handle, eenth | nineteen, seventeen, seventy |
| 6 | ensional, insk, ploy | M, M, m |
| 7 | allowance, \u2605, ther | ii, Bs, B |
| 8 | ylon, works, plays | EDITION, o\u011f, nt |
| 9 | ysc, oreal, Friend | B, K, B |
| 10 | redits, rossover, ameron | F, K, k |
| 11 | Tiger, urses, aught | N, W, C |
| 12 | aughter, gling, eland | L, l, L |
| 13 | othe, cano, ensity | S, s, S |
| 14 | ISTORY, hum, pots | H, H, h |
| 15 | gers, iegel, ki | S, s, S |
| 16 | ya, seq, est | selves, T, i |
| 17 | tl, ictionary, latch | R, R, D |
| 18 | Fres, pine, delay | R, u, llah |
| 19 | Shades, went, culosis | G, G, S |

Table E. 12: Left and Right Singular Vectors at Layer 11 - Head 5

| Rank | Top-3 Left Words | Top-3 Right Words |
|---|---|---|
| 0 | assail, challeng, achie | ertodd, \u25fc, \ufffd\u9192 |
| 1 | WARE, padding, req | \u9f8d\u5951\u58eb,    StreamerBot, soDeliveryDate |
| 2 | uing, anche, Inquis | heit, MpServer, partName |
| 3 | ward, ops, actory | builds, projects, Building |
| 4 | ary, bell, vis | ouf, unt, article |
| 5 | ments, Poo, emo | Will, Will, terday |
| 6 | abdom, book, Til | reads, read, writing |
| 7 | admission, Fighters, agy | model, Models, ilib |
| 8 | line, lines, se | line, lines, Hold |
| 9 | iness, less, ood | udic, ridden, usky |
| 10 | absence, inar, Miko | place, Must, must |
| 11 | hawk, nect, aff | esson, sees, scene |
| 12 | ie, een, ennett | Say, ighting, features |
| 13 | Peaks, construed, anguages | finding, find, Find |
| 14 | ming, mers, pling | ufact, Put, say |
| 15 | Authority, urated, disregard | record, records, Record |
| 16 | cript, Seen, Crash | Written, course, arium |
| 17 | ually, gladly, ously | tions, show, find |
| 18 | im, ading, Expand | image, Image, Image |
| 19 | NX, W, ees | swer, \u30c7\u30a3, report |

Table E. 13: Left and Right Singular Vectors at Layer 11 - Head 6

| Rank | Top-3 Left Words | Top-3 Right Words |
|------|------------------|-------------------|
| 0 | issue, txt, Princ | isSpecialOrderable, DragonMagazine, \ufffd\ufffd |
| 1 | mes, same, resa | guiActiveUn, Yanuk, Beir |
| 2 | eatured, avier, AMES | quickShipAvailable, BuyableInstoreAndOnline, RH |
| 3 | Levine, estone, Bronx | skirts, Els, Bris |
| 4 | lder, xit, Sav | Sov, grap, Al |
| 5 | xual, ss, soup | Orient, owship, toile |
| 6 | rices, glers, lishing | Uni, Tik, en |
| 7 | imation, hammer, nels | BAD, Ze, sa |
| 8 | saturated, lying, Past | Ry, AG, Val |
| 9 | activity, ozy, oko | Ay, AW, Ay |
| 10 | ows, aghan, ergy | Gul, cl, Nex |
| 11 | yrs, ish, hood | Wh, Har, Mart |
| 12 | omp, grandmother, MS | sidx, Alb, CTR |
| 13 | ses, ski, doctor | AD, ython, Ty |
| 14 | heed, Monthly, angan | OPS, Tur, Tam |
| 15 | Agency, VP, lex | Red, Grey, Redd |
| 16 | FORE, sil, hing | wcsstore, uci, Winged |
| 17 | idences, ining, ahl | Ste, Pend, hal |
| 18 | iance, taxpayers, anches | Fuj, appl, Zamb |
| 19 | ischer, apo, hiatus | Zamb, Zer, Nek |

Table E. 14: Left and Right Singular Vectors at Layer 11 - Head 7

| Rank | Top-3 Left Words | Top-3 Right Words |
|------|------------------|-------------------|
| 0 | shortest, ses, mentally | iHUD, DragonMagazine, Downloadha |
| 1 | our, ourselves, we | ourselves, ours, our |
| 2 | himself, lements, them | \u899a\u9192, natureconservancy, er-todd |
| 3 | etitive, EStream, workshop | FTWARE, SourceFile, \ufffd\u9192 |
| 4 | \u0627\u0644, holders, mileage | your, Free, Your |
| 5 | am, 'm, myself | my, myself, me |
| 6 | themselves, auder, ighthouse | Companies, theirs, THEIR |
| 7 | stract, hop, \u00a2 | soDeliveryDate, Civil, civilian |
| 8 | bage, ros, hyster | bage, aukee, Free |
| 9 | shop, acter, Shop | Humans, ourning, electronically |
| 10 | ¡+, myself, pse | my, myself, markets |
| 11 | Hold, SE, istant | ilage, roups, usra |
| 12 | uffs, VG, GG | verty, Leilan, Soft |
| 13 | sters, ual, ted | machine, machine, business |
| 14 | making, weights, mare | centrif, istani, culture |
| 15 | uador, oust, ertain | us, ours, our |
| 16 | vable, cam, ophy | system, System, systems |
| 17 | exch, velength, un | Games, abeth, gaming |
| 18 | latex, Edwards, Conway | Commercial, Community, community |
| 19 | ificial, rating, nces | ificial, System, technology |

Table E. 15: Left and Right Singular Vectors at Layer 11 - Head 8

| Rank | Top-3 Left Words | Top-3 Right Words |
|------|------------------|-------------------|
| 0 | the, in, a | \ufffd\ufffd\ufffd, guiActiveUn, cloneembedreportprint |
| 1 | *., ., determin | ,, the, - |
| 2 | and, ,, Un | arnaev, DragonMagazine, BuyableInstoreAndOnline |
| 3 | ?", ?), ?), | ?'", TPPStreamerBot, ',” |
| 4 | ,'", GIF, ," | Orderable, \ufffd, \ufffd |
| 5 | .'", They, .' | .'", '.", )." |
| 6 | ,', \u2010, ,'" | ,'", ',, ' |
| 7 | ,', ,'", ,' | ,'", ,', ,'" |
| 8 | .', ,', '. | .', ,', '. |
| 9 | her, she, She | she, hers, her |
| 10 | \ufffd, \ufffd, " | \ufffd, \ufffd, " |
| 11 | "..., ",", ", | )",, )",", "), |
| 12 | )."., ")., ..." | )."., ."[, ")." |
| 13 | us, )",, our | )",, )."., ,") |
| 14 | ));, );, ), | ",, ',, )); |
| 15 | ]., ];, ], | };, ];, '; |
| 16 | ..., ...", ... | ...], :], ..." |
| 17 | ?], !], .] | \u2026], !], ?] |
| 18 | ();, her, He | hers, ();, His |
| 19 | \u00ad, \u300f, You | \u300f, ¿., \u00ad |

Table E. 16: Left and Right Singular Vectors at Layer 11 - Head 9

| Rank | Top-3 Left Words | Top-3 Right Words |
|---|---|---|
| 0 | esthes, Eat, pts | DragonMagazine, Canaver, natureconservancy |
| 1 | ups, motors, hinted | confir, \ufffd, unlaw |
| 2 | pursue, pursuit, Frie | ticket, Desire, iferation |
| 3 | posted, dates, rece | achievement, unlocking, Hilbert |
| 4 | differential, prise, ushing | acceptance, handled, accepting |
| 5 | Hide, etsu, LET | optimizations, prioritize, emphasized |
| 6 | ously, uffer, ca | opsis, \u30df, stall |
| 7 | ann, Horn, Specifications | restraint, notice, surprises |
| 8 | supremacy, argon, ifier | ACTIONS, Contin, rue |
| 9 | ling, ceived, inf | errors, misunderstanding, accuracy |
| 10 | ittal, ampton, feld | denotes, denote, hazard |
| 11 | inf, andy, ery | plagiar, mentors, recommending |
| 12 | Soon, \ufffd, \ufffd | lax, Talks, Fell |
| 13 | cia, war, Fighters | dissatisf, consum, dissatisfaction |
| 14 | NAS, Schwar, Streamer | delet, sidx, inem |
| 15 | Glory, uan, ment | Reviewed, Congratulations, congratulations |
| 16 | frey, clay, essional | quirks, Integration, distinguishing |
| 17 | uck, marked, Request | appreciation, Guidelines, guidelines |
| 18 | prints, forcefully, Cel | conviction, convictions, impressions |
| 19 | utic, endez, inging | disag, bruising, spo |

Table E. 17: Left and Right Singular Vectors at Layer 11 - Head 10

| Rank | Top-3 Left Words | Top-3 Right Words |
|------|------------------|-------------------|
| 0 | above, former, dm | \u25fc, Downloadha, Canaver |
| 1 | Cohen, oku, Corporation | be, ache, the |
| 2 | liar, Ross, Irving | Rossi, Mind, Zen |
| 3 | Treatment, MT, tubing | etts, Taylor, Tan |
| 4 | Torch, dt, Honour | Divinity, marqu, vine |
| 5 | ==, Sinn, imitation | Stafford, Bradford, Halo |
| 6 | asks, fitted, caution | BW, BW, Berger |
| 7 | encer, hero, success | Gon, Johnny, PATH |
| 8 | Chung, anke, IRE | Chennai, Carey, Carmen |
| 9 | Commodore, iom, attract | curry, Cunningham, clam |
| 10 | earth, CS, oyal | Sov, Trin, paralle |
| 11 | ramid, el, DIT | Hilton, diarr, \ufffd\u9192 |
| 12 | ulla, alde, uality | McInt, alde, Idle |
| 13 | cam, write, ports | Cave, Chal, Connie |
| 14 | buf, anne, Emin | Dwar, Dwarf, Das |
| 15 | job, play, job | buquerque, Liber, reb |
| 16 | ASC, ector, Order | Sorceress, Alic, Astro |
| 17 | ting, enced, te | Forest, Kan, tree |
| 18 | ater, Turner, UAL | \u9f8d\ufffd, Omn, Gamma |
| 19 | Matrix, RIP, oping | Fed, STEP, Rand |

Table E. 18: Left and Right Singular Vectors at Layer 11 - Head 11

| Rank | Top-3 Left Words | Top-3 Right Words |
|------|------------------|-------------------|
| 0 | 8, 9, 6 | \u899a\u9192, cloneembedreportprint, StreamerBot |
| 1 | ", [?], , | DragonMagazine, cloneembedreportprint, ertodd |
| 2 | puff, rem, Ey | \ufffd\ufffd\u58eb, catentry, Flavoring |
| 3 | air, compressor, exchange | air, blow, nose |
| 4 | burn, burning, burns | burns, burning, burn |
| 5 | smoke, blowing, sky | smoke, clouds, airflow |
| 6 | light, shade, lighting | light, illumination, Light |
| 7 | break, breaks, Bre | breaks, breaker, broken |
| 8 | finger, air, registrations | finger, finger, Feet |
| 9 | rolls, roll, rolled | Rolls, rolls, Ludwig |
| 10 | opening, opened, closing | opened, closes, opening |
| 11 | ause, blank, generating | rawdownloadcloneembedreportprint, ause, sburg |
| 12 | anne, \u0639, sprayed | \u30fc\ufffd, \ufffd\ufffd, iltration |
| 13 | ear, audio, Ear | ear, ears, Ear |
| 14 | goggles, watched, devotion | ideos, TPS, goggles |
| 15 | leaf, slashed, hunger | gou, ouri, margins |
| 16 | voices, voic, Hand | voic, leash, voiced |
| 17 | short, Short, short | shorten, shortened, short |
| 18 | tones, tones, tone | bells, tone, marrow |
| 19 | drawn, connected, ieties | wu, River, Awakening |

