# OpenReview forum: "Traveling Words: A Geometric Interpretation of Transformers"
_ICLR.cc/2024/Conference — Submitted to ICLR 2024_

### Official Review · Reviewer_zeL2 · 2023-10-31

**Soundness:** 2 fair
**Presentation:** 2 fair
**Contribution:** 2 fair
**Rating:** 5
**Confidence:** 2

**Summary:**

This is a typical outlier paper. The submission firstly proposes an interpretation that the layer normalzation operation projects an embedding to a subspace perpendicular to [1,1,...,1] then normalizes it to a hypersphere in the subspace. Then the weight matrices in a transformer shift the projected hyperspere's position and shapes. Then the authors probe the embeddings of common words using cosine similarity. The conclusions are: (1) early layers slightly shift the embeddings to their typical contexts; (2) late layers are not understandable; (3) one case shows that the shifted embeddings are closer to the last word in a sentence.

**Strengths:**

+ This is a creative and interesting *OUTLIER* paper. Some of these papers can be quite influential.
+ I am a computer vision person and as far as I know the geometric interpretation of LayerNorm is new.

**Weaknesses:**

- The experiments are only case studies in a small scale. And I cannot say the conclusions are meaningful or not. What's worse, some analysis does not lead to conclusions, e.g., later layer embeddings are not understandable.
- To be honest, I fail to get why the word travelling analysis is related to the geometric interpretation of LayerNorm. We can still do these analyses without the d-1 projection interpretation right? Correct me if I am wrong.

**Questions:**

See the last box.

---

### Official Review · Reviewer_qGqz · 2023-11-01

**Soundness:** 3 good
**Presentation:** 3 good
**Contribution:** 2 fair
**Rating:** 3
**Confidence:** 4

**Summary:**

This paper explores geometric views into some of the primitive operations in Transformer architectures. The starting point is layer normalization which projects and normalizes a vector so that it will lie on a hypersphere. Then the query-key matrix implements an affine transformation bringing closer on the hypersphere terms that are related. The value-output matrix can be seen as an additional key-value store in the attention layer, while the computation of output probabilities reflects the similarity of the final layer represention as projected on the (shared) hypersphere with the embedding vectors in the vocabulary.

(Probing) experiments quantify these observations using a pre-trained GPT-2 model. The impact of layer normalization on word embeddings is verified, query-key transformations seem to exhibit interpretable patterns at the first layer and some key-value heads work together at the last layer to preserve input key meanings. Similarly some patterns could be captured in singular vector pairs of the key-value and query-key matrices (the latter at layer 0). This work concludes with the representation of the second to last token in a sentence which can be seen that gets closer and closer to the last token representation (in a projected view) as the layers of the network model are traversed.

**Strengths:**

- This is an easy to follow paper, with interesting and intuitive geometric arguments, supported by simple matrix formulas.

- Some of the examples/demonstrations reveal patterns which tend to happen either on early or deep layers and could loosely fit into the high-level geometric insights developed.

**Weaknesses:**

- The novelty of this work is limited since the key observations have already been mentioned in other works (that are adequately cited): [Brody et al.] for layer normalization, query-key matrix; [Millidge & Black] for value-output matrix.

- The "journey" of the representation of one word towards the representation of the next one in a sentence is interesting but is could well be an artifact of the reduction in dimensionality in the projection as also noted. Regarding the examples that are expected to frame the geometric arguments presented, they are either very slim in volume to be conclusive (when some signal/pattern is observed) or there is simply no interesting pattern that could be easily extracted.

**Questions:**

- Regarding the traveling words interpretation: It would be nice to test it with more sentences and alternatively work with the original encoding vectors through the layers (no reduction: do the distances to the last token representation decrease? what about the respective distances for successive tokens not  at the end of a sentence? is there a intuitive way to argue for a possible pattern in this?)

---

### Official Review · Reviewer_TdFy · 2023-11-01

**Soundness:** 2 fair
**Presentation:** 2 fair
**Contribution:** 2 fair
**Rating:** 3
**Confidence:** 2

**Summary:**

The paper aims to interpret the mechanisms of transformers and establishes an explanation for the effect of layer normalization from a geometric viewpoint. The interpretation is validated via probing a GPT-2 model.

**Strengths:**

* The paper provides an intuitive, geometry perspective for interpreting the Transformer architecture.
* Empirical probing experiments on GPT-2 validated some claims in the paper.

**Weaknesses:**

* Some ideas discussed in the paper, such as interpreting LayerNorm as surface projection have been discussed in prior works and are not novel. A discussion on the novelty of the proposed paper and how it compares with prior works will help clarify this concern.
* The paper provides an interesting perspective on the specific architecture in popular implementations of Transformers, but its applications or insights for further results are not fully discussed in the paper.

**Questions:**

* Figure 1 and Figure 4 suggest that work particles travel along the path determined by residual updates, but such a description is very general. Are there more specific properties within the residual updates?

---

### Official Review · Reviewer_rASQ · 2023-11-04

**Soundness:** 1 poor
**Presentation:** 1 poor
**Contribution:** 2 fair
**Rating:** 3
**Confidence:** 4

**Summary:**

In this paper, the authors introduce a novel geometric perspective to shed light on the inner workings of transformers. Their main contribution is the findings on how layer normalization constrains the latent features of transformers to a hyper-sphere, which in turn allows attention mechanisms to shape the semantic representation of words on this surface. This geometric viewpoint connects various established properties of transformers, including iterative refinement and contextual embeddings. To validate their insights, the authors analyze a pre-trained GPT-2 model with 124 million parameters. Their findings unveil distinct query-key attention patterns in early layers and confirm prior observations about the specialization of attention heads in deeper layers. By leveraging these geometric insights, the paper offers an intuitive understanding of transformers, depicting iterative refinement as a process that models the trajectory of word particles along the surface of a hyper-sphere.

**Strengths:**

1. The paper presents a study on understanding transformers through the lens of layer normalization, a key component in transformers, and the matrices $W_{QK}, W_{VO}$ used in the attention mechanism.


2. The main insights are that in each layer, the layer normalization projects the features to a shared hyper-sphere. The proposed interpretation of attention is similar to the feed-forward module by Geva et al. (2021) in that both calculate relevance scores and aggregate sub-updates for the residual stream. However, the key difference lies in how scores and updates are computed: attention relies on dynamic context, while the feed-forward module depends on static representations.

3. The authors validate these insights by probing a pre-trained 124M parameter GPT-2 mode

**Weaknesses:**

1. The presentation can be improved significantly. I find it hard to see the differences from prior works and what exactly are the main contributions of this paper.

2. Most of the emprical results are using some selected examples and I do not quite follow these results. Could you list the main points that you are making from these experiments and how the evidence justifies them?  What is the trajectory in figure 4 trying to show?

**Questions:**

In A.1  $\mu$? is the average of components of a feature vector $x_i$? Can you provide clear definitions of what the features, mean, and std. deviation are? I would imagine $\mathbf{\mu}$ to be either an expectation of $\mathbf{x}$ or an average of $\mathbf{x}_i$.

---

### Author Response · Authors · 2023-11-18
**Discussion Regarding Prior Work**

We would like to thank the reviewers for their time and effort on evaluating our paper, and want to take this opportunity to provide clarification on how the proposed interpretation differs from prior work cited in the paper.

As mentioned by the reviewers, Brody et al. were the first to introduce the geometric interpretation of LayerNorm and its relationship to vector projection and scaling. Nonetheless, their main focus was on how this interpretation allows for the computability of certain functions within a single attention block (specifically the _majority_ function), a goal the authors explicitly refer to in the title of their paper as the "Expressivity Role of LayerNorm".

We extend this interpretation to the entire transformer architecture by analyzing its consequences on the internal representation space of every layer:

* First, we slightly modify the decomposition proposed by Brody et al. to derive a formulation of LayerNorm using only the geometric primitives (vector projection and scaling), such that orthogonality to the $\overrightarrow{\mathbf{1}}$ vector does not arise as a numerical consequence of subtracting the mean (as discussed in Brody et al.), but is instead proposed as an intrinsic property of the normalization step. To illustrate, note that under our formulation (Equations 3 & 4) is easy to see how this projection can be generalized to vector/hyperplane combinations different from $\frac{1}{\sqrt{d}}\overrightarrow{\mathbf{1}}$, while it is not straightforward to do so under Brody et al.'s interpretation.

Equiped with a strong geometric intuition on what LayerNorm does, we revisit the ideas of Xiong et al., Geva et al. and Millidge & Black:

* [Xiong et al.] We discuss at the end of Section 2.1 how the arrangement of transformer layers proposed by Xiong et al. (commonly known as Pre-LN Transformers) affects the model's internal representations, such that **all attention and feed forward layers within a transformer share the same input representation space**, which is different from the representation space of the residual stream, although it is related to it via projection onto the $d-1$ hypersphere (defined by $\frac{1}{\sqrt{d}}\overrightarrow{\mathbf{1}}$). This is a notable result, as common intuition would suggest that the representation space of each layer after normalization would be unique, given that the mean and standard deviation are data-dependent variables. Under the proposed geometric interpretation of LayerNorm is easy to see this is not the case, as all layers apply exactly the same projection to the representation space of the residual stream.

* [Geva et al. and Millidge & Black] In Section 2.2.2 we discuss how the SVD method proposed by Millidge & Black can be interpreted in relation to the key-value stores introduced by Geva et al. In both of these works,  linear probing has been limited to the value "side" of either the VO matrix or the feed forward layer, that is, the parameter groups that directly interact with the residual stream. From Millidge & Black: _"Our method is extremely simple. Take a weight matrix M of the network. Take the SVD of this matrix to obtain left and right singular vectors M=USV. Take whichever matrix has the same dimensionality as the residual stream (typically the right singular matrix V)"_. Similarly, Geva et al. only probe the weights of the MLP value matrices, and rely on trigger examples (instead of probing) to interpret the information encoded in the MLP key matrices. We complement their work by **introducing a probing methodology to understand both sides of these key-value matrices** in Section 3.3.1 and show the results in Appendix E.

---

### Meta-Review · Area_Chair_zjCo · 2023-12-15

**Metareview:**

The paper attempts to improve our understanding of the transformers. In this regards, the authors introduces a geometric perspective on the internal mechanisms of transformers using the key idea that layer normalization confines latent features to a hyper-sphere. Empirical experiments are conducted using a pre-trained GPT-2 model. The reviews raised several concerns and the author's response did not change any of the reviewers opinion. In particular, reviewers point out that the presentation needs improvement, the novelty compared to prior work is unclear, and the empirical results are not entirely convincing. Unfortunately, the general consensus among reviewers is towards the paper not being strong enough for acceptance, with suggestions for clarifying its contributions and novelty in comparison to existing works.

**Justification For Why Not Higher Score:**

- Presentation and Clarity: All the reviewers expressed concerns about the presentation of the paper, including why "traveling" in the title is relevant to the geometric interpretation.
- Novelty and Contribution: There were doubts regarding the novelty of the paper's contributions, particularly in comparison to existing works in the field. The paper needs to more clearly delineate how its approach and findings differ from and advance beyond what is already known.
- Empirical Validation: The empirical results presented in the paper were not entirely convincing to the reviewers.

**Justification For Why Not Lower Score:**

N/A

---

### Decision · Program_Chairs · 2024-01-16

Reject